# Deconstructing body axis morphogenesis in zebrafish embryos using robot-assisted tissue micromanipulation

Ece Özelçi [1,2], Erik Mailand [1], Matthias Rüegg[1], Andrew C. Oates [2] ✉ & Mahmut Selman Sakar [1,2] ✉

Classic microsurgical techniques, such as those used in the early 1900s by Mangold and Spemann, have been instrumental in advancing our understanding of embryonic development. However, these techniques are highly specialized, leading to issues of inter-operator variability. Here we introduce a user-friendly robotic microsurgery platform that allows precise mechanical manipulation of soft tissues in zebrafish embryos. Using our platform, we reproducibly targeted precise regions of tail explants, and quantified the response in real-time by following notochord and presomitic mesoderm (PSM) morphogenesis and segmentation clock dynamics during vertebrate anteroposterior axis elongation. We find an extension force generated through the posterior notochord that is strong enough to buckle the structure. Our data suggest that this force generates a unidirectional notochord extension towards the tailbud because PSM tissue around the posterior notochord does not let it slide anteriorly. These results complement existing biomechanical models of axis elongation, revealing a critical coupling between the posterior notochord, the tailbud, and the PSM, and show that somite patterning is robust against structural perturbations.

Developing embryos display remarkable spatiotemporal coordination of tissue morphogenesis and patterning. To explore this fundamental aspect of development, experimentalists draw on a diverse set of tools, ranging from genetics to pharmacology to mechanical perturbation. However, discoveries continue to be made using classical techniques, in use for more than a century, to physically manipulate the embryo. These techniques allow microsurgical de- and reconstruction of the embryo, particularly in larger embryos that are easy to handle such as tunicates, sea urchins, flatworms, chick and amphibians[1–5]. The key operations in experimental embryology are: create defects; isolate individual parts; recombine an embryo's parts in the wrong spatial location; and transplant parts from one embryo to another, often in different locations and at different times[5]. Post-operation, the embryo, or part thereof, is allowed to develop and any resulting defects in the developmental sequence are used to infer causal processes. Although

techniques now include laser ablation[6–8] and genetically induced cell death[9,10], experimental embryology remains largely manual and requires expertise. Furthermore, despite some model organisms offering potential advantages in genetics or imaging, embryo size also remains prohibitive.

 Robotic surgery holds the potential to broaden the applicability of traditional tissue manipulation techniques and improve an experimentalist's access to them. Importantly, robotic surgery allows amplitude scaling of hand movements, reduction of tremor, and teleoperation[11,12]. Robotic assistance has enabled surgeons to perform complex procedures with more precision, flexibility, and control[13]. Inspired by this success, here we develop a robotic platform to aid manipulation of small embryos without prolonged training. As a proof of principle example, we investigate the contribution of different embryonic tissues to anteroposterior (AP) axis elongation and segmentation in

[1]Institute of Mechanical Engineering, Ecole Polytechnique Fédérale de Lausanne (EPFL), 1015 Lausanne, Switzerland. [2]Institute of Bioengineering, EPFL, 1015 Lausanne, Switzerland. ✉e-mail: andrew.oates@epfl.ch; selman.sakar@epfl.ch

developing zebrafish embryos, while reading out real-time dynamics of tissue morphogenesis and segmentation gene expression.

The biomechanics of vertebrate tail elongation have been described in terms of forces applied by the axial tissues (i.e., the notochord and the neural tube), and forces generated within the tailbud and the presomitic mesoderm (PSM). According to one theory[14], tail elongation is driven by an engine-like mechanism where advancing axial tissues (which includes the notochord) push posteriorly against the tailbud. This force would displace tailbud cells such that they move anteriorly into the PSM that sits bilaterally to the axial structures. In turn, expansion of the PSM compresses the axial tissues, promoting their elongation in a feedback loop. Another theory proposes a compaction-extension mechanism in which surrounding tissues compress the PSM and the tailbud to drive the elongation[15]. In an alternative mechanism, unidirectional axis extension has been explained by a jamming transition from a fluid-like behaviour in the tailbud to a solid-like behaviour in the posterior PSM[16]. Cells entering the tailbud from the dorsal-medial region cause the expansion of the fluid-like tissue, with the solid-like PSM acting as a rigid support that biases tissue expansion towards the posterior direction. However, it is still an outstanding question whether the proposed models are mutually exclusive or complementary.

AP axis elongation is spatiotemporally coordinated with the rhythmic and sequential segmentation of PSM into tissue blocks called somites in the anterior-most PSM. The segmentation clock consists of a population of cellular oscillators in the tailbud and PSM that control somite formation[17,18]. Patterning of the PSM has been studied by quantifying tissue level kinematic waves of gene expression from transgenic reporters of the segmentation clock[17–20]. Although the majority of studies have focused on genetics and biochemistry, recent reports have investigated biomechanical influences on somite boundary formation and the oscillations of the segmentation clock[21,22]. How the biomechanics of elongation impact the segmentation clock or somitogenesis remains unclear.

In this paper, we use robot-assisted micromanipulation to carry out a series of operations on the tail of the elongating and segmenting zebrafish embryo. In combination with time-lapse microscopy, this allows us to investigate existing hypotheses about the biomechanics of elongation and its potential effects on somitogenesis with precision, repeatability, and ease.

## Results

We developed a compact ($200 \times 100 \times 70 \, \text{mm}^3$), high resolution (4 nm position and $25 \, \mu°$ rotation), and dexterous (6 degrees of freedom) robotic microsurgery system (Fig. 1a, Supplementary Note 1). The system is modular, designed to accommodate a variety of instruments useful in embryology including actuated microtools (tweezers, scissors) and electrocautery device (Fig. 1b). The use of the micromanipulator and end-effector increases dexterity in composing translational and rotational movements, eliminates the tremor of the operator's hand, and allows precise adjustment of the tool's position and movement. To streamline robot-assisted operations and subsequent time-lapse imaging, we fabricated agarose plates with arrays of wells from 3D printed moulds (Supplementary Fig. 1). Aiming to reproducibly target specific AP locations, use of a microscope-integrated robotic microsurgery platform with precise placement of the microscissors allowed us to operate on specimens with required accuracy (Supplementary Fig. 2).

### Tail explants maintain coordinated elongation and segmentation

To investigate whether elongation of the zebrafish tail depends on the anterior part of the trunk (from head to cloaca[23]), we first isolated the tail as an explant (from somite 11 to the end of the tailbud), from a 15-somite stage zebrafish embryo. Removal of the tail can be done manually with forceps and a microknife or microscissors, but in the hands of an inexperienced experimentalist, this operation typically leads to irregularly sized and/or damaged explants. To avoid this, we teleoperated microscissors mounted on the robot for microsurgical operations, which helped us to repeatedly produce viable tail explants with controlled size (Supplementary Note 2). Robot-assisted microsurgery is significantly faster and more accurate compared to manual surgery in performing microsurgical cuts (Supplementary Fig. 3).

We used three quantitative metrics to evaluate tail elongation and tissue morphogenesis: (1) the rate of explant elongation, (2) the period of morphological somite formation, and (3) the period of the segmentation clock (Fig. 1c–f). We primarily used bright-field images to quantify elongation and the period of somite formation. To quantify the period of the segmentation clock we traced the signal intensity of an oscillatory clock reporter gene (Her1-YFP) with fluorescent microscopy either with an upright compound microscope or with a light-sheet microscope. Light-sheet microscopy was also used to count the number or observe the shape of individual cells inside the nuclear-marked Histone (H2B-mCherry) transgenic embryos, using its high time and z-axis spatial resolution.

After obtaining the tail explant with robot-assisted microsurgery (Fig. 1g) we observed normal somite and notochord morphology (Fig. 1h). Lateral and dorsal views of the tail explant via light sheet fluorescence microscopy showed that anatomical arrangements of the tissues, and the left-right symmetry in the segmentation process were preserved (Fig. 1h). During the course of elongation, periodic somite formation continued in the explants while the tail gradually straightened as in intact embryos (Fig. 1i, Supplementary Movie 1). The tail explants contained $20 \pm 1$ somites at the end of somitogenesis ($N = 4$, $n = 12$). Including the 10 somites that were left in the anterior part of the embryo, the total number of somites is $30 \pm 1$. This result is consistent with measurements performed in intact embryos ($30 \pm 1$, $N = 3$, $n = 30$) and reported in the literature[24–26]. The somites at the anterior end of the tail explant displayed the classic chevron shape (Fig. 1i), suggesting that the tail is not only elongating, but coordinating with the development of the musculoskeletal system. Tail explants started to twitch before the completion of somitogenesis, showing another hallmark of proper neuro-muscular development (Supplementary Movie 2). Taken together, these data suggest that the tail explant has sufficient information and resources to coordinate tissue morphogenesis and patterning/segmentation.

We next asked whether the dynamics of elongation and somite formation are affected by the dissection. The tail explants elongated with a constant rate during the course of three hours following the surgical operation, as was also evident in intact embryos (Fig. 2a). However, there was a decrease in the average elongation rate of tail explants by 18% ($0.7 \pm 0.06 \, \mu\text{m min}^{-1}$) when compared to intact embryos ($0.9 \pm 0.06 \, \mu\text{m min}^{-1}$) (Fig. 2b). We consistently started with tail explants containing five formed somites, but we found elongation rate to be independent of the initial somite number (Supplementary Fig. 4). This developmental slowing is consistent with previous in vitro cultures of brain explants[27] and individual segmentation clock cells[28] as well as trunk explants from zebrafish[29], suggesting that the experimental perturbation and/or absence of the anterior structures alter overall growth rate in the explant.

In addition to the reduced elongation rate, the period of somite formation was also slower in the explant ($33 \pm 2.5 \, \text{min}$) compared to the intact embryo ($28 \pm 2 \, \text{min}$) to the same degree (i.e., 16%) (Fig. 2c). However, the final AP length of the tail explants was comparable to that of intact embryos at the same developmental stage (Supplementary Fig. 5a). As expected from these observations, the AP somite length in the explant was almost identical to the intact embryo (Supplementary Fig. 5b, c). Thus, slowing of both elongation and somitogenesis in the tail explant indicates that coordination between these two processes remains intact.

Consistent with slower morphological development, segmentation clock oscillations in the anterior PSM were also slower, with the average period of oscillation 19% longer in the tail explants (34 ± 2 min) than in the intact embryo (28.5 ± 3 min) (Fig. 2d). This change in dynamics was consistent throughout the measured cycles (Fig. 2e, Supplementary Fig. 6). This data suggests that the increase in segmentation clock period and the decrease in elongation rate are coordinated to produce somites with the same length.

Altogether, these results show that robot-assisted microsurgery is an efficient and reliable method to produce explants suitable for spatiotemporal analysis of morphological and patterning events during development.

## Somite formation does not require tail elongation

Our observations of coordinated elongation and segmentation dynamics suggested that these processes may be dependent

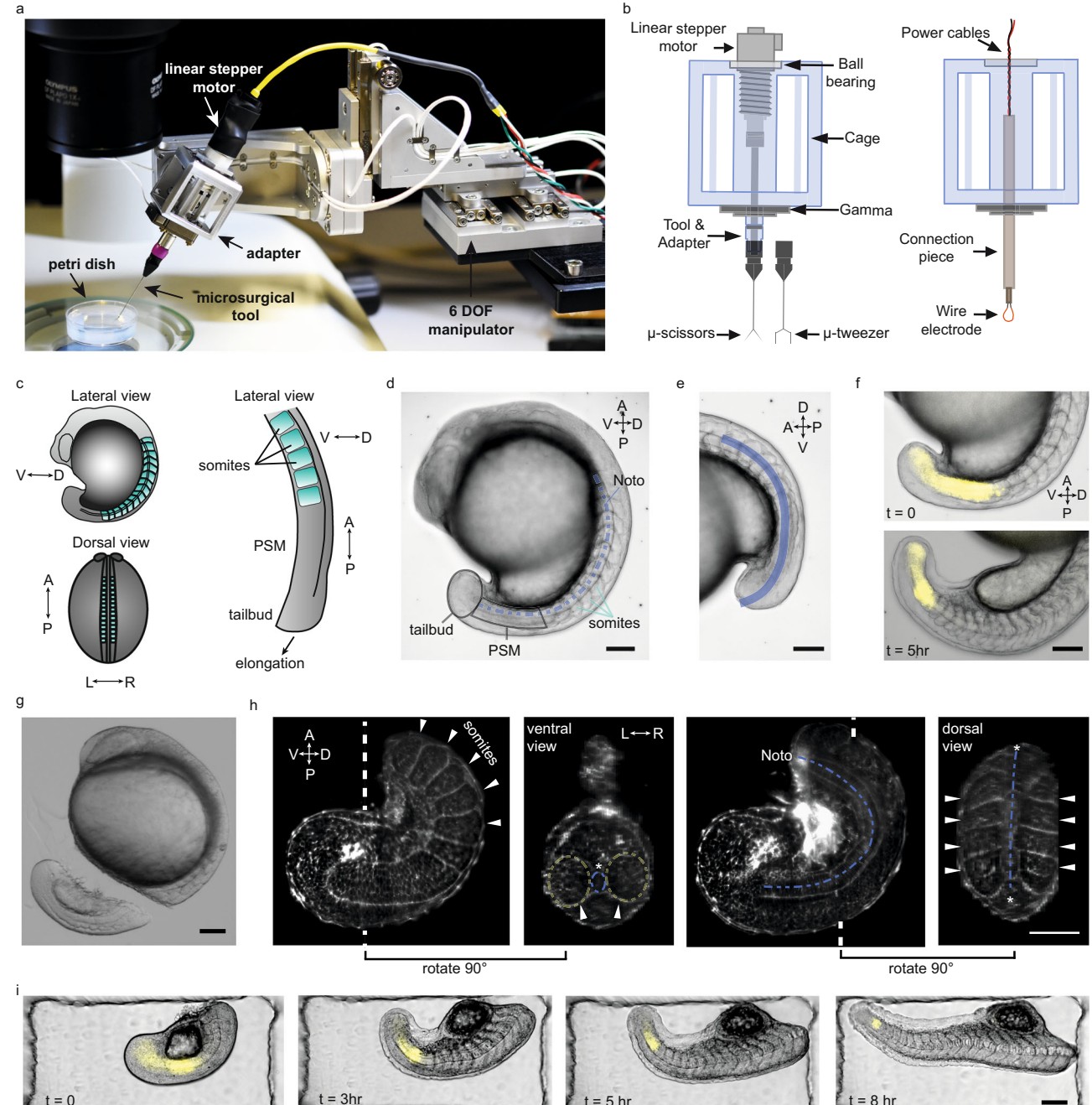

**Fig. 1 | Robotic platform enables precise microsurgery of the zebrafish tail. a** Robotic tissue micromanipulation platform along with the stereo microscope and operation chamber. **b** Schematic illustration of the adapters designed to hold actuated and non-actuated instruments (not to scale). **c** Schematic showing the zebrafish embryo from different anatomical axes (V/D: ventral/dorsal, A/P: anterior/posterior, L/R: left/right). **d** A representative bright field (BF) image of a zebrafish embryo. Tissues that are studied in this work are indicated on the embryo. **e** Line of interest indicated with blue is generated to measure the AP tail length from BF image shown in (**d**). **f** Composite images of the embryo showing BF and Her1-YFP channels at different time points. **g** A BF image of the embryo right after robot-assisted microsurgery. **h** Light-sheet fluorescence image of a tail explant from a utr-mCherry transgenic line which marks filamentous actin structures. White dashed lines indicate the plane at which ventral and dorsal-view images were taken. White arrows indicate the somites, blue dashed-lines indicate notochord (Noto: notochord). **i** Composite images of a tail explant over time showing the elongation of the tail along with Her1-YFP signal. Scale bars, 100 μm.

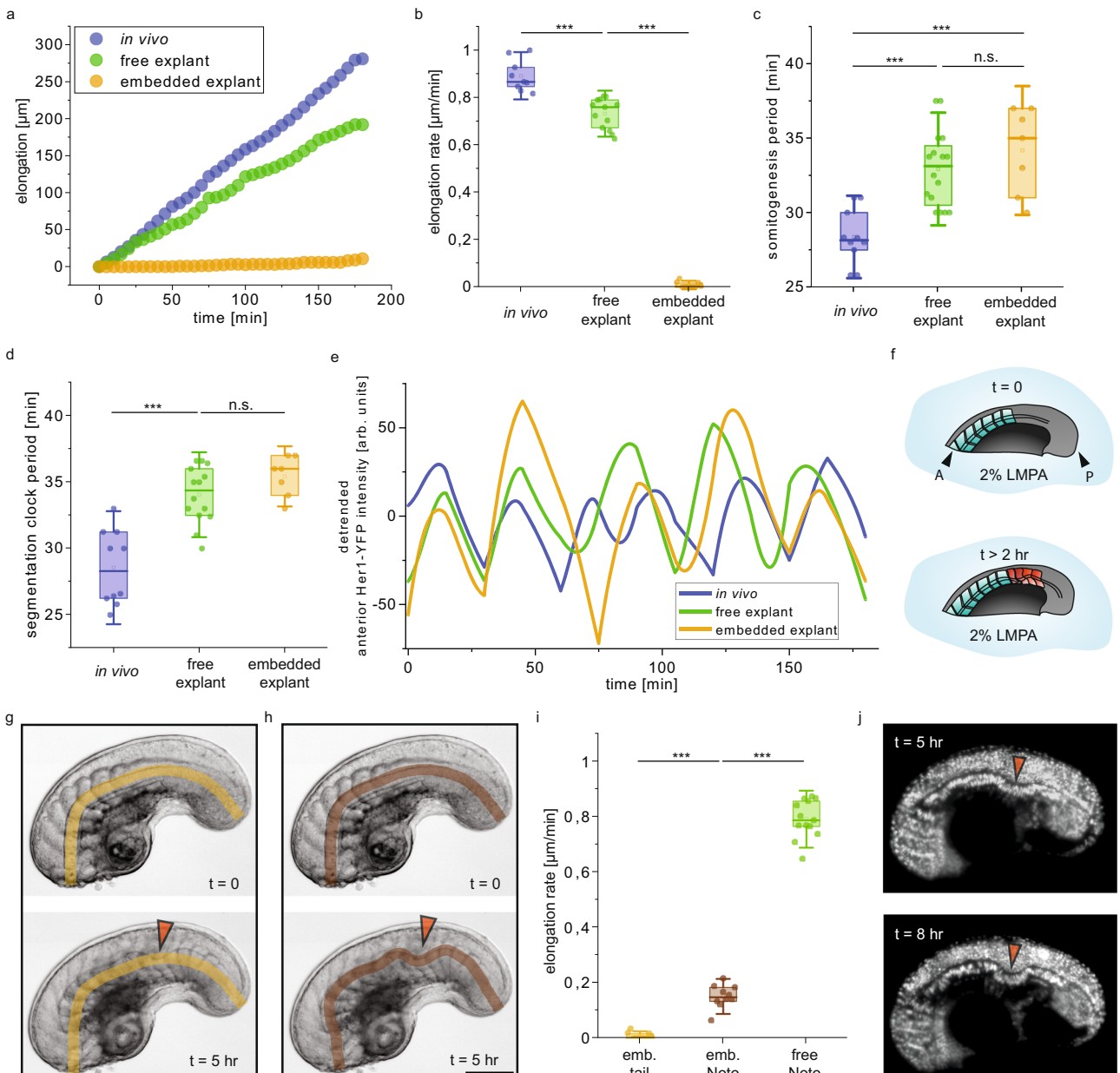

**Fig. 2 | Embedding of explants dissociates notochord elongation and tail morphogenesis. a** Representative elongation vs time curves for in vivo embryos, and explants that are free to elongate or embedded inside agarose gel. **b** Elongation rates extracted from elongation vs time curves, such as shown in (**a**) ($N = 4$, $n = 9$ embryos, $N = 4$, $n = 13$ free explants, $N = 3$, $n = 10$ embedded explants). **c** Average period of somitogenesis in intact embryos ($N = 4$, $n = 10$), free tail explants ($N = 4$, $n = 16$), and embedded tail explants ($N = 2$, $n = 7$). **d** Average period of Her1-YFP oscillations in intact embryos ($N = 4$, $n = 10$), free tail explants ($N = 4$, $n = 14$), and embedded tail explants ($N = 2$, $n = 7$). **e** Representative plots showing the intensity of Her1-YFP signal over time. The signal is recorded in the anterior part of the intact embryos, free tail explant and embedded tail explants. **f** Schematic showing the tail explant embedded in 2% LMPA (low melting point agarose). Embedding leads to structural defects in the somites and the notochord (A: anterior, P: posterior).

**g, h** Representative time-lapse BF images of a tail explant under confinement. Orange arrows indicate the location of buckling of the notochord. (g) shows the length measurement based on the anterior and posterior ends of the tail explant, termed tail elongation. **h** The length measurement based on the notochord of the tail explant, termed notochord elongation. **i** Elongation rates of the embedded tail explant (emb. tail) ($N = 3$, $n = 10$), notochord of the embedded tail explant (emb. Noto) ($N = 3$, $n = 10$), and the elongation rate of the notochord of the free explant (free Noto) ($N = 4$, $n = 13$). **j** Representative time-lapse fluorescence images of tail explants expressing H2B-mCherry. Orange arrows indicate the location and pro- gression of buckling of the notochord. Scale bars, 100 μm. Box plots in panels (**b**), (**c**), (**d**) and (**i**) indicate the median (mid-line), 25th and 75th percentiles (box), and 1.5× the interquartile range (whiskers). All statistical comparisons are performed with one way ANOVA test, ***$P < 0.001$, n.s. indicates not significant.

upon each other. To investigate this, we followed somite forma- tion in explants embedded inside 2% (w/w) agarose gel (Fig. 2f), which physically constrains the explant and prevents elongation. As expected, the length of the tail explant did not increase (Fig. 2a, b). Somitogenesis continued in the embedded explants with a period matching the free explants (Fig. 2c) however, somite length was shortened along the AP axis, and PSM length

was strongly reduced (Fig. 2g, Supplementary Fig. 7). Similar observations were made previously using compression between two cover slips to block elongation of a dissected trunk explant[29]. Despite the change in somite length of the embedded explant, the period of the segmentation clock in the anterior PSM matched that of the free explant (Fig. 2d, e), suggesting that patterning may be decoupled from elongation.

Shorter somite AP length was accompanied by an elongation of the DV length (Supplementary Fig. 7), with an increased internal chevron angle (Fig. 2g, Supplementary Fig. 7). This DV lengthening appeared to compensate for the loss of AP length as the number of cells in the newly formed somites did not differ significantly between embedded and free tail explants (Supplementary Fig. 8), suggesting that overall somite volume remained unchanged[22]. Both the somite deformation and PSM shortening indicate that cells are abnormally rearranged and distributed in the embedded explants. In this respect, this experimental manipulation is similar to the classical cell recombination experiments done by Holtfreter with *Xenopus* cells[30]. Taken together, these data show that the period of the segmentation clock can be decoupled from the biomechanics of elongation.

### Notochord elongation does not require tail elongation

Time-lapse images of the embedded explants revealed that the notochord was elongating, and we noticed other active morphological changes. Notably, the notochord buckled in all the embedded tail explants (Fig. 2g) with the degree of bending increasing over time (Fig. 2h, Supplementary Figs. 9, 10). Increased bending indicated a continued elongation of the notochord despite a lack of tail outgrowth. However, elongation rate of the notochord in embedded explants was significantly smaller than the free explants (Fig. 2i). This can be explained by not having enough space to elongate in the embedded configuration. The direction of buckling informs about the distribution of stresses on the notochord. The notochord buckled ventrally in all embedded explants, at an AP position coinciding with the location of the first somite that formed after the tail explant had been embedded. The consistent buckling location indicated the existence of a localized mechanical instability in the AP axes of the tail explant. The site of notochord buckling, which started after the last somite on the tail explant, shifted anteriorly with $0.1 \pm 0.02\ \mu m\ min^{-1}$ ($N = 2$, $n = 6$) (Fig. 2j, Supplementary Movie 3). Later, the notochord buckled a second time at a location posterior to the first (Supplementary Fig. 9a). Given that force applied anteriorly in the notochord could not transmit posteriorly across the initial buckle, this second buckle indicates forces originating in the posterior notochord. Combined, we show that when the notochord elongates not in tandem with the tail, it generates enough force from the posterior to buckle. This result suggests that coordination of notochord and tail elongation protects the notochord from mechanical defects. It also raises the possibility that the axial stresses generated by the posterior notochord may play a important role in the axial elongation of the entire tail.

### Tail elongation is primarily coordinated by the posterior tissues

We next investigated how morphogenesis is spatially coordinated throughout the elongating tail. To test the contribution of the parts of the notochord located in the anterior versus those in the posterior PSM region to the elongation process, we first dissected tail explants into two smaller pieces using robot-assisted microsurgery. Using the microscissors, we generated an anterior piece (explant-A), which contained anterior notochord, already formed somites and the anterior half of the PSM, and a posterior piece (explant-P), containing posterior notochord, the posterior PSM half and the tailbud (Fig. 3a). The high-resolution motion of the robotic manipulator, together with the advantage of working at higher magnification without the problem of having tools in the field of view, enabled repeated and precise execution of this operation in these smaller samples. Explant-A showed minimal change in AP length while that of explant-P approximately doubled over 3 h (Fig. 3b). The combined AP length as well as the elongation rate of the explants (explant-A plus explant-P) was comparable to that of intact tail explants at 3 h post-dissection (Fig. 3c). This indicates that the forces responsible for notochord and tail elongation are localized almost entirely within the posterior half of the tail explant, and that biomechanical interactions between the

anterior and posterior regions of the explant do not play a measurable role.

The separation of the tail explant into two parts also enabled us to explore whether signals and forces generated within the anterior region (including anterior part of the notochord, somites and PSM) would influence segmentation in the posterior PSM, and vice versa. Consistent with observations in avian embryos[31,32], somites only formed in explant-P once explant-A had finished segmenting, mimicking the order of somite formation in the intact tail explant (Fig. 3b). The length and shapes of the somites formed in explant-A and explant-P over 5 h were also similar to intact tail explants (Supplementary Fig. 11). Previous work using fixed samples has shown that anterior waves of *her1* transcription are not abolished by the removal of the zebrafish tailbud[33]. Consistent with these results, explant-A and explant-P retained their Her1-YFP wavefront schedule, completing oscillations and segmentation in explant-A and thereafter in explant-P (Fig. 3d). The spatiotemporally coordinated oscillations seen in the physically isolated explant pieces suggest that cells inside the PSM have the necessary information in their local environment to become somites in the right place and with the correct morphology.

Given that we observed elongation activity mainly occurs in explant-P, we sought to define the smallest part of the posterior tail that can sustain elongation by carrying out a series of dissections creating successively smaller posterior explant-Ps. We found that explant-Ps smaller than 150 μm in length (measured from the tailbud end) did not contain an observable notochord and did not elongate, instead contracting into a sphere (Supplementary Fig. 12, Supplementary Movie 4). Thus, internal processes within these explants, which consist mainly of tailbud, are not sufficient to drive elongation.

### Biomechanical coupling among posterior tissues in the elongating tail

Our results suggest that the forces involved in tail elongation are generated posteriorly, but not primarily within the posterior of the tailbud. The engine-like mechanism[14] of elongation proposed in the chick embryo involves the interplay of tissues along the AP axis including the notochord, posterior PSM, and tailbud. To systematically investigate the biomechanical interactions among the tailbud, notochord, and PSM in the posterior of the tail explant, we performed targeted mechanical perturbations by creating defects in various locations.

To test whether the notochord drives tail elongation, and whether or not notochord elongation is in turn driven by the surrounding tissues, we first ablated the posterior notochord using an image-guided UV laser microsurgery platform. UV laser ablation allowed for targeted local damage of cells in the desired part of the tail explant. The platform allows rapid light-sheet scanning of the samples, and ablation of prescribed 3D volumes with cellular resolution within seconds (Supplementary Fig. 13). We ablated the notochord starting from its posterior end to -100 μm anterior (Fig. 4a). This ablation resulted in an interruption to tail elongation for $68 \pm 15$ min after which elongation re-started (Fig. 4b) with a 26% reduced rate compared to the unablated explant of $0.54 \pm 0.1\ \mu m\ min^{-1}$. Some of the somites that formed next to the ablated notochord region had abnormal shapes (Fig. 4c, Supplementary Fig. 14), suggesting that neighbouring PSM may have been damaged by the ablation. Importantly, the notochord continued to elongate during the stall in tail elongation by the addition of cells from the progenitor domain in the tailbud immediately posterior to the ablated region (Fig. 4d).

We next ablated the posterior PSM and observed the elongation. Damage to both left and right sides of the posterior PSM (Fig. 4e) on the same tail explant also caused a transient stall in tail elongation (Fig. 4f). Analogous to the results of the notochord ablation experiments, tail elongation ceased for $111 \pm 30$ min then resumed with a 37% reduced rate of $0.46 \pm 0.1\ \mu m\ min^{-1}$ (Fig. 4g). Again, despite the stall in tail elongation for ~2 h following the ablation, the notochord

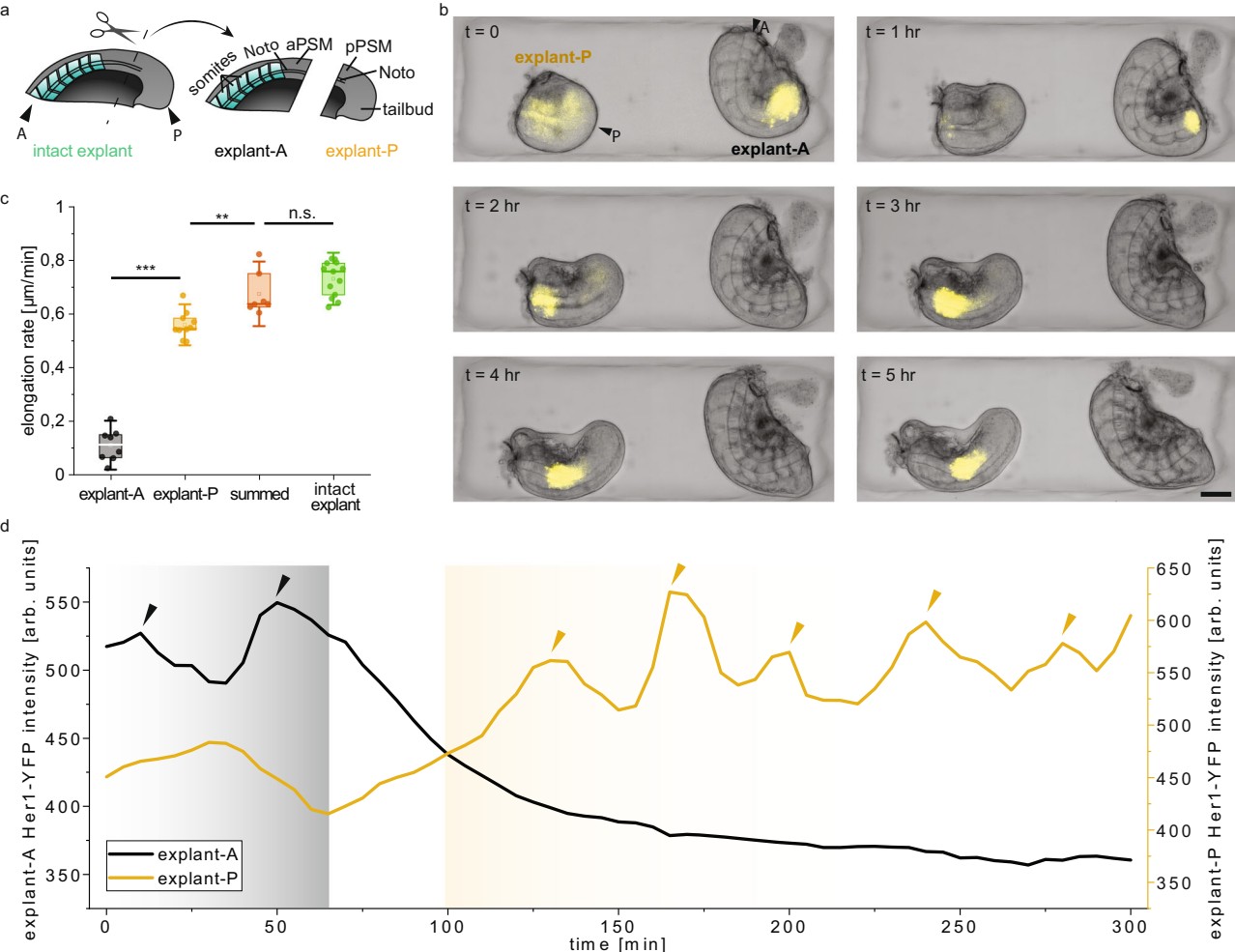

**Fig. 3 | Dynamics of elongation and signalling in the anterior and posterior parts of the tail explant are independent. a** Schematic showing the dissection of the tail explant into an anterior (explant-A) and a posterior (explant-P) piece. Right schematic shows the main structures we focus on in this paper for explant-A and -P (A: anterior, P: posterior, Noto: notochord, a/pPSM: anterior/posterior PSM). **b** Composite image of BF and YFP channels showing the elongation and Her1-YFP expression in explant-A and explant-P cut from the same tail explant. A and P are placed on the top left image to indicate the anterior and posterior of the entire explant. Scale bar, 100 μm. **c** Elongation rates of the explant-A ($N = 4$, $n = 8$), explant-P ($N = 4$, $n = 10$), the sum of the elongation rates of explant-A and explant-P (summed) ($N = 4$, $n = 7$) and the intact tail explant ($N = 4$, $n = 13$). Box plots indicate the median (mid-line), 25th and 75th percentiles (box), and 1.5× the interquartile range (whiskers). Statistical comparisons are performed with One Way ANOVA test for which ***$P < 0.001$, **$P < 0.01$, n.s. indicates not significant. $P = 0.002$ for the comparison between explant-P and summed. **d** Intensity of Her1-YFP signal in the anterior part of explant-A and explant-P. Grey and yellow shaded regions respectively indicate the time interval where the anterior and posterior explant pieces are segmenting.

continued to elongate throughout this period (Supplementary Fig. 15, Supplementary Movie 5). Somitogenesis was severely disrupted in the ablated region, as expected, due to the destruction of most PSM cells however, somites continued to form in the elongating part of the tail posterior to the ablation as PSM progenitor cells moved out of the tailbud (Supplementary Fig. 15, Supplementary Movie 6). Newly formed somites gradually pushed anteriorly into the ablated region, reducing its size throughout the observation. Thus, ablation of either posterior notochord or posterior PSM is enough to temporarily stop elongation, suggesting that an interaction between the two is required to drive tail elongation.

We hypothesize that the resumption of tail elongation after the stall depends on the presence of progenitors in the intact tailbud, immediately posterior to the ablated region. As a first step to test this idea, we removed the entire tailbud from the tail explant, starting from the end of the notochord to the posterior-most tip of the tail (Fig. 5a). Removal of the entire tailbud dramatically reduced elongation of the remaining explant by 86% ($0.1 \pm 0.05\ \mu m\ min^{-1}$), including that of the notochord. Consistent with our observations in explant-A (Fig. 3b, c),

somites formed with the expected AP length in the pre-existing PSM (Supplementary Fig. 16a). These results demonstrate that the tailbud is not necessary for ongoing somitogenesis, but is critical for the elongation of all tail tissues.

To determine if this elongation activity could be further spatially localised within the tailbud, we created defects in specific tailbud regions. Our microscissors could not generate clean cuts required for the intended defects, so we instead used a robotically assisted electrocautery device (Fig. 1b). Previous work has shown that during normal development, cells are added to the tip of the notochord from a progenitor domain that resides within the tailbud immediately posterior the notochord[34]. To determine if the severe disruption to elongation upon tailbud removal was due to general perturbation of tailbud integrity, we removed the posterior-most tip of the tailbud while preserving the notochord progenitor domain. Use of the robot-assisted electrocautery device allowed us to remove a small piece from the tip of the tailbud without damaging the rest of the tail explant. The tail explant continued to elongate with a 22% reduced rate ($0.58 \pm 0.1\ \mu m\ min^{-1}$) compared to the unoperated explant. We

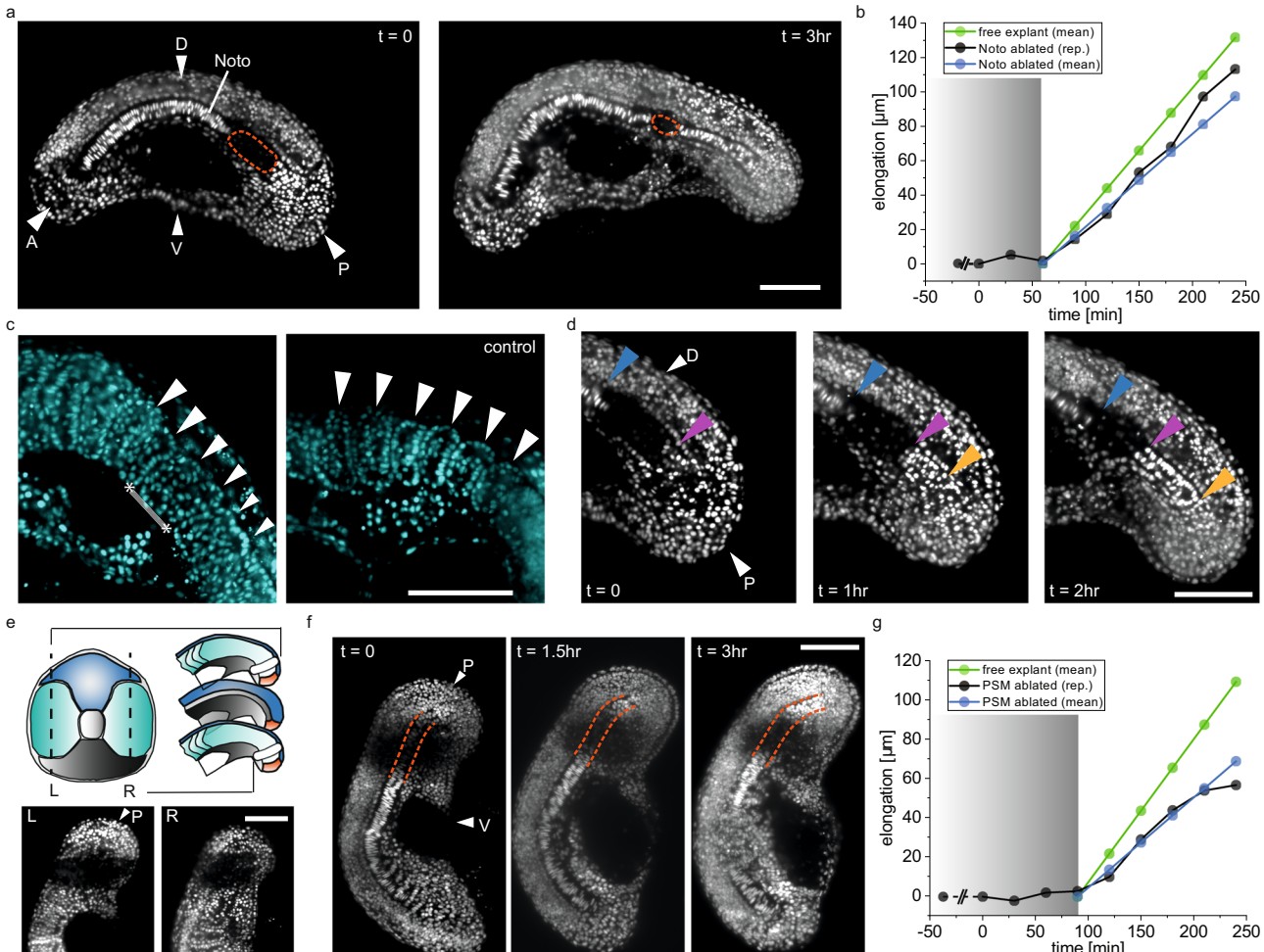

**Fig. 4 | Laser ablation of the posterior notochord and PSM reveals interaction required for tail elongation. a** Time-lapse fluorescence images of a H2B-mCherry tail explant in which the posterior end of the notochord is ablated using laser microsurgery. Ablated region is indicated with the orange dashed line (Noto: notochord, A/P: anterior/posterior, D/V: dorsal/ventral). **b** The elongation of the tail over time for the explant shown in (a). Grey shaded part indicates the period during which the elongation process is affected. Time passed from ablation until the start of the movie is indicated before $t = 0$ with dashed line (free explant $N = 4$, $n = 13$, Noto ablated explant $N = 5$, $n = 10$). **c** Somites formed next to the ablated notochord region (highlighted with the line in between asterisks) are malformed. White arrows indicate the somites. **d** Time-lapse fluorescence images of the ablated notochord region show the elongation of the notochord at the tip. Blue, purple and yellow arrows indicate the anterior and posterior ends of the ablated region and the posterior most end of the elongating notochord respectively. **e** (top) Schematic showing the right and left part of the tail explant that are ablated from the mediolateral and anteroposterior views (L/R: left/right, P: posterior). (bottom) Fluorescence images showing the ablated regions of both sides of the PSM. **f** Time-lapse fluorescence images of a tail explant where both sides of the posterior PSM are ablated. Orange dashed lines are to see the notochord easily. **g** Tail elongation curve of the explant shown in (f). Grey shaded part indicates the period during which the elongation process is affected. Time passed from ablation until the start of the movie is indicated before $t = 0$ with dashed line (free explant $N = 4$, $n = 13$, PSM ablated explant $N = 5$, $n = 11$). Scale bars, 100 μm.

consistently observed that the notochord elongated towards the ventral side (Fig. 5b). Somites were smaller in the dorsoventral axis, which could be explained by the limited number of progenitor cells in the reduced tailbud (Supplementary Fig. 16b).

To test if the notochord progenitor domain is responsible for the elongation of the tail, we ablated this region using the UV laser microsurgery platform. Despite the rest of the tissue remaining intact, ablation of the progenitor domain in a diameter of ~60 μm reduced the elongation of the tail explant by 48% ($0.38 \pm 0.1$ μm min$^{-1}$), with the posterior end of the notochord always bending ventrally within the tail (Fig. 5c, Supplementary Fig. 17a). Ablating the same volume of tissue (Supplementary Fig. 18) within the tailbud in a region outside the progenitor domain had a lesser impact, reducing elongation by 29% ($0.52 \pm 0.1$ μm min$^{-1}$) (Fig. 5d, Supplementary Fig. 17b). This demonstrates the large role played by the notochord progenitors in tail elongation. The data associated with the tailbud manipulations are presented in Fig. 5e.

Together our experimental embryology approach suggests that elongation of the notochord relies on the progenitor domain, and that the proper elongation of the tail along the AP axis is maintained by interaction of the elongating notochord with the tailbud and the PSM.

## Discussion

Here we have described a micromanipulation toolkit that brings classical embryology to zebrafish embryos in a more precise and reproducible way. While classical embryology is primarily qualitative, our platform enables quantitative characterization of cellular-level responses to tissue-scale perturbations. At the centre of the toolkit lies a dexterous microrobot that is capable of manipulating instruments that are widely used to operate on embryos, as well as actuated medical tools such as micro scissors. Custom-design sample holders facilitated both microsurgical operations and medium-throughput time-lapse imaging. Fine mechanical manipulation of the zebrafish embryo would normally require extensive training and experience

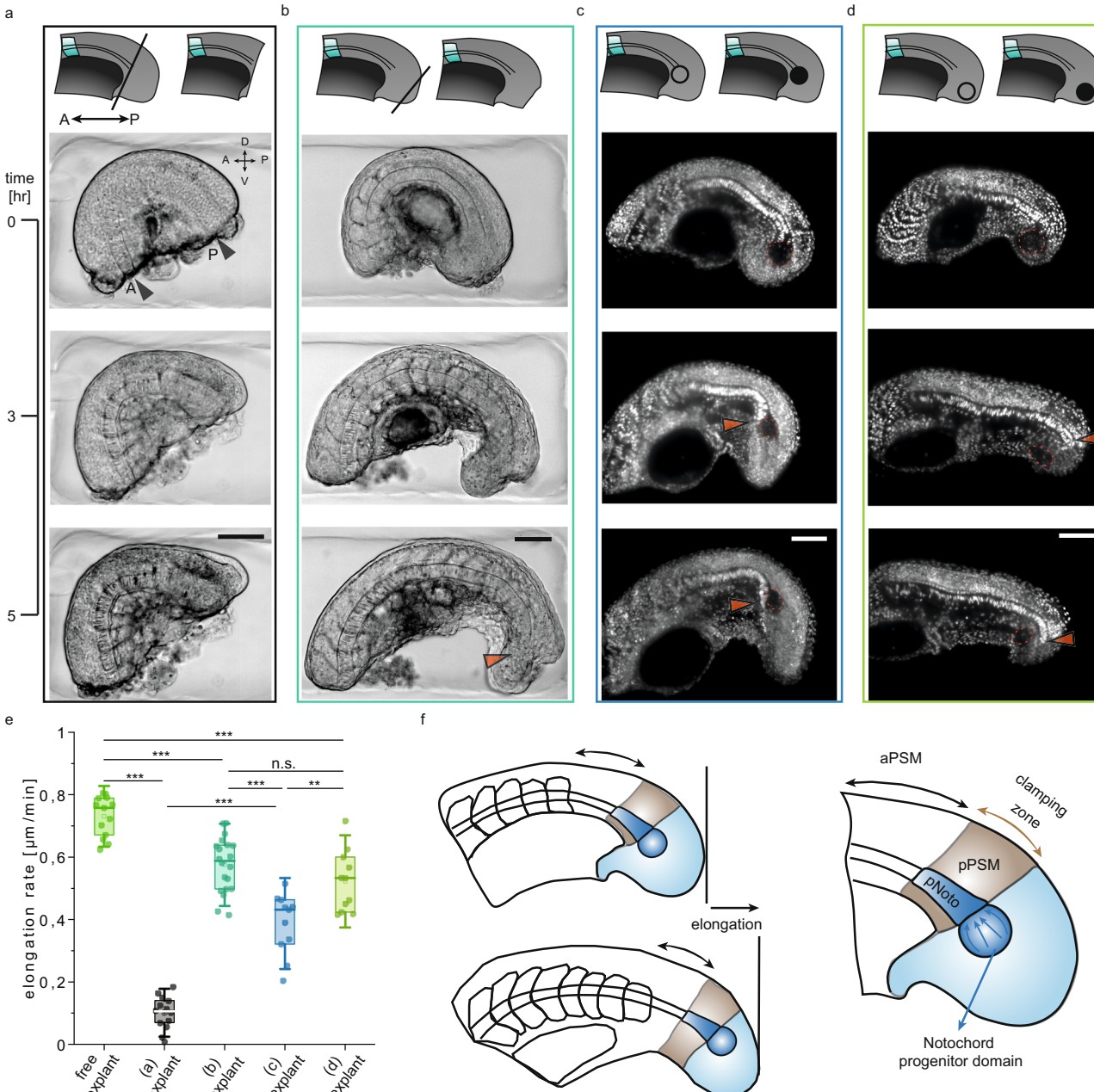

**Fig. 5 | Micromanipulations to the tailbud show that cell addition to the end of the notochord from the notochord progenitor domain is required for tail elongation.** Schematics describing the micromanipulations are shown at the top of each case. **a** Time-lapse BF images of a tail explant where the tail bud is removed using the microcautery device (A/P: anterior/posterior, D/V: dorsal/ventral). **b** Time-lapse BF images of a tail explant where the tip of the tail bud is removed using the electrocautery device. Orange arrow shows the location towards which the notochord elongates. The notochord progenitor domain is maintained intact. **c** Time-lapse fluorescence images of a H2B-mCherry labelled tail explant where the notochord progenitor domain is ablated using laser microsurgery. Orange arrow shows the location where the notochord bends towards the ventral side of the tail.

**d** Time-lapse fluorescence images of a H2B-mCherry labelled tail explant where the tip of the tailbud is ablated. **e** Elongation rate of the free explants ($N = 4$, $n = 13$) and explant samples presented in (**a**) ($N = 4$, $n = 14$), **b** ($N = 6$, $n = 20$), **c** ($N = 5$, $n = 11$), **d** ($N = 5$, $n = 11$). Box plots indicate the median (mid-line), 25th and 75th percentiles (box), and 1.5× the interquartile range (whiskers). Statistical comparisons are performed with one way ANOVA test for which ***$P < 0.001$, **$P < 0.01$, n.s. indicates not significant. $P = 0.004$ for the comparison between (**c**) explant and (**d**) explant. **f** Mechanism of the tail elongation in observed stages. Key players are named and coloured ((p)Noto: (posterior)notochord, a/pPSM: anterior/posterior PSM). Clamping zone refers to the posterior part of the tail where the notochord is held in position. Scale bars, 100 μm.

however, our robot-assisted microsurgery platform does not require prior expertise and is compatible with automation, making mechanical manipulation on small and delicate specimens such as the zebrafish embryo accessible to the broader biophysics community. The mesoscale tissue manipulation techniques presented in this work seamlessly complement local micromanipulation techniques based on embedded magnetic nanoparticles[35] and microdroplets[36], which

together have the potential to reveal biomechanical principles of morphogenesis.

Our results from the posterior PSM ablation experiments suggest that notochord elongation in the zebrafish tail is not driven by convergent extension, as previously suggested for avian embryos[14], but driven by cell addition from the progenitor domain. In the jamming transition theory[16,37], the PSM acts as a solid support and prevents the

posterior fluid-like part of the PSM and tailbud from being overtaken by surface stresses. There is no clear role for the notochord in this scenario. Nevertheless, our experiments show that the notochord can exert significant forces along the axis, and that continuous elongation of the notochord is essential for proper tail morphogenesis. We propose that the unidirectional elongation of the notochord along with the tail depends on the posterior region of the notochord to be held in position by the posterior PSM (Fig. 5f). Thus, we showed here that tail elongation at the observed developmental stage is driven by coupled interactions between three units: the notochord progenitor domain, the posterior notochord, and the posterior PSM.

Linking our state-of-the-art experimental embryology results to existing biomechanical theories allows us to speculate that the forces holding the posterior notochord in place during elongation could be generated by three mechanisms: friction, compression, and interlocking due to a phase transition. As reported for universal soft grippers, these are not mutually exclusive mechanisms, and could operate in combination[38]. Friction through contact mechanics could prevent the notochord from slipping anteriorly, similar to previous ideas that PSM provides resistance to the extensive forces generated during vacuolation of the maturing notochord[34]. It is also possible that PSM may be expanding due to the material flow from the tailbud as suggested for avian embryos[14], pressing against the notochord. Finally, a transition from an unjammed, deformable state to a jammed state with solid-like rigidity, as previously suggested for zebrafish embryos[16,37], may generate interlocking. Small modifications of the packing density in the vicinity of the jamming transition can drive dramatic changes in the mechanical response[39], a feature that has been harnessed to engineer universal grippers[40].

The morphology and structure of the *no tail* and *her1*[+/-]*her7*[+/-] mutant embryo lines provide independent support to our observations on the relatively weak coupling between somitogenesis and elongation. *no tail* mutants do not have a mature notochord posterior to the trunk, yet they maintain somitogenesis in a dramatically reduced tail structure, with the somites appearing compressed in the AP axis and fusing across the axial midline[41–43]. In contrast, *her1*[+/-]*her7*[+/-] mutants have an intact notochord and elongate with a rate comparable to control embryos (see Supplementary Fig. 19), but they do not form proper somite boundaries[44]. These genetic observations are in accordance with our microsurgery results showing that elongation of the tail does not depend on the segmentation of the PSM, and that the presence of a notochord is required for tail elongation.

Tail explants completed elongation and somitogenesis at reduced rates that were nonetheless balanced such that the segment lengths in the explant were indistinguishable from those formed in the intact embryo. This result is similar to embryos that are grown at lower temperatures[45]. Temperature-induced change in somitogenesis period and axial elongation rate are compensated across a wide range of temperatures, consequently the somite length distribution remains the same. Temperature has a direct effect on the rates of biochemical reactions, as does altering the flux of metabolic energy. Recent experiments investigating the species-specific difference in segmentation clock period in mouse (2 h) and human (6 h) using in vitro systems composed of PSM-like pluripotent stem cells from each species have implicated differences in metabolic rate as a key influence. Expression and degradation rates of *HES7*, a homolog of *her1* in zebrafish, are slower in human cells than in mouse cells[46]. Moreover, mass-specific oxygen consumption rate and glycolytic proton efflux rate were twice as slow in human cells than in mouse cells[47]. Notably, reducing these metabolic rates by inhibiting the electron transport chain slowed down the segmentation clock. It is conceivable that generating the explant reduced the metabolic rate in the isolated tail tissue, perhaps by reducing the quantity of yolk, the metabolic energy source used by the embryo, that is associated with the cells of the PSM and tailbud. Future work can test whether increasing metabolic rate in the explant will restore a normal developmental rate.

As discussed above, in the freely elongating tail explant, or the embryo[45], the period of segmentation is balanced with the elongation rate. However, although the period of the segmentation clock is not affected by embedding, tail elongation is effectively stopped. This shows that the period of the clock can be uncoupled from the elongation rate of the tail. In most models of segmentation, the readout of the clock in the anterior PSM is controlled by FGF and Wnt signal gradients that extend anteriorly from the site of synthesis in the tailbud[48–53]. The posterior regression of the tailbud in the extending body axis is thus thought to create a co-moving wavefront of signal activity that defines a position in the tissue where the clock's timing is used to determine the segment length. However, the embedding results clearly show that an ongoing posterior regression of signal gradients is not required for the formation of new segments, nor is it required for their correct timing, at least over the 3 h of our time-lapse observations. This suggests that the behaviour of segmentation clock cells in the anterior PSM is unlikely to be directly controlled by a threshold—or some other simple feature—of a moving signal gradient[29]. Nevertheless, it leaves open the possibility that changes in signal levels in the posterior PSM may play a role, potentially by causing changes to the cells leaving the tailbud that become visible with some substantial time delay[53].

Microsurgical manipulation of the chick embryo was critical in establishing the concept of the segmentation clock. Pioneering experiments indicated that *c-hairy1* gene expression waves continued on schedule in the anterior of the PSM, at least for one cycle, when the posterior was removed, thereby ruling out the propagation of a signal from the posterior as the cause of the wave pattern. We have now repeated and extended these experiments in a transgenic zebrafish embryo carrying a *her1*-based reporter of the segmentation clock, showing in real-time and over multiple cycles that there is a developmental schedule of oscillations across the tissue that operates without the requirement for ongoing input from the anterior or posterior of the oscillating tissue. These findings support the idea that the wave patterns are locally autonomous to the tissues. While recent studies using cultures of reconstituted mouse PSM tissues[21,54] and organoids of pluripotent human stem cells[55,56] suggest that wave patterns may self-organize, single cells dissected from the tailbud or PSM of the zebrafish embryo showed Her1-YFP oscillations very similar to the cells residing in the embryo[19], suggesting that the observed tissue autonomy of the embryonic pattern is a direct result of an autonomous pattern at the cellular level.

The micromanipulation tools that are introduced in this study enable the precise and reliable mechanical perturbation of the local microenvironment in living embryos, as well as the removal of precisely sized and shaped explants. Analysis of these operations can help to answer outstanding questions regarding whether body axis morphogenesis, or other accessible patterning processes, are best described as the sum of cell-autonomous events, or as phenomena that emerge collectively at the multicellular level.

In summary, we leveraged the precision and dexterity offered by robotic micromanipulation techniques to systematically perturb the physical structure of the developing zebrafish embryo. The presented methods complement the growing arsenal of mechanical characterization tools that are tailored for in vivo measurements.

## Methods

### Zebrafish care
Fish were maintained following standard protocols. Embryos are obtained by natural spawning. Her1-YFP[20], H2B-mCherry[57], utr-GFP[58] transgenic lines are used from EPFL (Lausanne, CH) fish facility. Both heterozygous and homozygous fish were used. All transgenic fish (H2B-mCherry, utr-GFP) were outcrossed with *her1* transgenic fish to

obtain animals with two different markers. After fertilization, embryos were kept at 28 °C until the shield stage, and then at 19 °C until 6–8 somite stage. Manipulations were performed at room temperature and imaging was performed at 28 °C. General animal experimental license of EPFL, granted by the Service de la Consommation et des Affaires Vétérinaires of the canton of Vaud – Switzerland (authorization number VD-H23), covers the experimental procedures.

## Gel confinement

Low-gelling temperature agarose (Sigma-Aldrich Chemie Gmbh, A9414-5G) was mixed with fresh fish water to prepare 2% low-melting point agarose (LMPA). This solution was put on explants covered with Leibovitz's L15 medium (ThermoFisher, Catalog No. 21083027) in a glass bottomed petri dish (35 mm). LMPA was cooled down to 24 °C to solidify the gel and embed the tail explants. After solidification of LMPA solution, 2 ml of L15 medium with Pen-Strep (P/S) (1:200 v/v) and tricaine (1:33 v/v) was added to the petri dish to prevent the sample from drying. After 8–12 h of confinement, explants taken from the agarose with the help of forceps and transferred to another plate filled with clean L15 medium.

## Fabrication of explant chambers

A 3D printer (Form 2, Formlabs) was used to manufacture the microsurgery and imaging chambers (Supplementary Fig. 1). STL files for the molds used to make the microsurgery and imaging chambers can be found in Supplementary Software 1. Resolution of the printer was set to 50 μm and clear resin was used. Liquid agarose solution (2% LMPA) was poured into a 35 mm glass-bottom petri dish, and the mould was gently pressed upon the pre-gel solution. The mould was gently removed from the agar solution after solidification for minimum of 20 min at 4 °C. For the light sheet imaging 3D printed mould of Viventis LS1 microscope was prepared as described in Supplementary Fig. 20.

## Robotic micromanipulation

We designed a 6-DOF robot with nested piezoelectric actuators that occupies a volume of 200 × 100 × 70 mm³. The design and operation of the system was inspired by an ophthalmic microsurgery platform[59,60]. We developed micromachined aluminium adapters that allow mounting of actuated tools such as scissors and tweezers along with non-actuated tools such as dissection knifes, needles, and glass capillaries. The robotic manipulation system is constructed from piezoelectric stick-slip actuators (Smartact GmbH) that have sub-micron positioning precision. It has three translation stages (X (SmarAct SLC-2460-D-L-E), Y (SmarAct SLC-2460-O-W-D-L-E), Z (SmarAct SLC-2460-D-L-E)), two rotary actuators equipped with positional feedback (α (roll) SmarAct SR-4513-D-S, β (pitch) SmarAct SR-2812-S) and a light-weight rotary stage without sensor (γ) in a 'T(X)-T(Y)-T(Z)-R(α)-R(β)-R(γ)'-configuration. To hold and actuate microsurgery tools, a custom designed adapter, mounted on β, incorporates the open-loop γ-stage for infinite axial rotation of the end-effector (Supplementary Fig. 21a). The actuation of tools such as microscissors was based on the movement of a plunger against a passive spring-loaded mechanism. The movement of the plunger was coupled to the translation of a metal tube over the blades, controlling the opening and closing of the tool. We added a transmission to improve the accuracy and reduce play, a crucial upgrade for reducing tremor and increase throughput. An external cage houses a pivot-mounted tubular motor support that was rotated by the γ-stage. The operation of the motor (Can-Stack 15000 series LC1574W-12-999, Haydon Kerk) is controlled by a programmable single-board Arduino microcontroller (UNO). The motor can be quickly removed for the installation of non-actuated tools. We developed a specific tool adapter for glass capillaries connected to a microinjection system (PV830, World Precision Instruments). The adapter design allows the use of different capillary sizes with tubing connection. The control software of the robot is developed within the micro-manager (μManager) framework[61]. Operation of the platform is demonstrated in Supplementary Movie 7.

## Microsurgery

Embryos were cut to small pieces using either a microknife (Fine Science Tools, Item No. 10318-14, Needle Blade) or surgical microscissors (Advanced DSP Tip 27+ Straight Scissors, 727.53, Alcon) mounted on the robot. The tail explants were kept for 30 min to 1 h before imaging to give enough time for healing of the skin. Subsequently, they were transferred to the imaging chamber. Embryos were also surgically manipulated with a microcautery device (Protech International Inc., MC-2010) using a 13 μm-thick wire electrode. The electrode was mounted on the robot.

## Laser ablation microsurgery

Pulsed UV laser (PNV-M02510-1×0, Teem Photonics) mounted to a Viventis LS1 light-sheet scope was used for all the laser microsurgeries. The peak power (75 mW) and the number of pulses applied in a given length (40 pulses per μm) were kept the same. A representative ablation process can be found in Supplementary Fig. 13a.

## Light-sheet imaging

Light-sheet imaging was performed with LS1 Live dual illumination light-sheet microscope (Viventis Microscopy Sarl, Switzerland). Up to 9 explants can be imaged within the same imaging chamber. The microscope was equipped with two 25X objectives (CFI75 Apochromat NA 1.1, Nikon) and three laser sources (488, 561, and 638 nm). Fluorescence excitation was performed with dual beam illumination light-sheet microscopy with 2.2 μm beam thickness. Signals were collected through a Nikon CF175 Apo LWD 25x/1.1 NA objective and 561/25 and 525/50-25 nm bandpass filters with a sCMOS camera (Andor Zyla 4.2 Plus). The pixel size was 0.3467 μm. Time-lapse movies were recorded for 5–8 h with 3 min time interval, 3 μm z-distance, and varying number of planes in the z direction depending on the thickness of the sample. 3D views of explants were constructed using a commercial imaging software (Imaris, Bitplane).

## Widefield imaging

Brightfield and YFP channels were imaged with a motorized inverted microscope (Nikon Ti-Eclipse). YFP and bright field signals were collected through a Nikon Plan Fluor 10× objective and 515 nm bandpass filter with a CMOS camera (ORCA-Flash4.0, Hamamatsu). Depending on the thickness of the samples, 3–5 z-slices with 20 μm steps were recorded.

## Image processing

Open-source software FiJi[62] was used to analyse the time-lapse bright field and fluorescent images. Elongation measurements were done manually using a line of interest width of 20 pixels. The bright field images were post-processed using the Gaussian-based Stack Focuser in Timelaps plugin with a blur of 2. *her1* signal intensity measurements were done from the max projection of YFP z-stacks. The average reporter intensity was measured using LOI interpreter in Timelaps plugin with a diameter of 40 pixels. The obtained mean reporter intensity measurements were transferred to OriginPro 2020b (Originlab) to quantify the anterior oscillation period over time. For the cell counting several FiJi[62] plugins were used. First, necessary files for the analysis were obtained with BigDataViewer[63]. Then, Mastodon was used to detect all the cells in the sample of interest. Later on, the somite of interest is selected in 3D with Labkit (Supplementary Fig. 8a, b). Finally, Paleontologist (https://github.com/bercowskya/paleontologist), a cell counting and visualization pipeline developed by Arianne Bercowsky Rama, was used to obtain the number of cells in the selected somite (Supplementary Fig. 8c).

## Statistical analysis

Measurements were done with distinct samples (n) from multiple independent biological replicates (N). Data processing was done using the native functions of Originlab. All the numbers reported for multiple samples are in the format of "mean ± standard deviation". Coefficient of variance for anterior somitogenesis and morphogenesis were done by dividing the standard deviation by the mean. In all the box plots lower and upper edges of the box represent the 25th and 75th percentiles, the mid-line represents the median and whiskers are extended to 1.5× the interquartile range. For the significance analysis One Way ANOVA test by using Tukey's multiple comparison was performed with *, **, *** corresponding $p < 0.05$, $p < 0.01$ and $p < 0.001$ respectively.

## Reporting summary

Further information on research design is available in the Nature Portfolio Reporting Summary linked to this article.

## Data availability

Data supporting the findings of this study are available within the paper and its Supplementary Information files. All other relevant data are available from authors upon request. Source data are provided with this paper.

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

## Acknowledgements

This work was supported by the EPFL SV iPhD program and the European Research Council (ERC) under the European Union's Horizon 2020 research and innovation program (Grant agreement No. 714609). We thank Can Aztekin, Laurel Rohde, Sundar Ram Naganathan, Arianne Bercowsky Rama and Olivier Venzin for fruitful discussions. We thank the zebrafish facility in EPFL, especially Chloé Jollivet, Florian Lang and Guillaume Valentin for their assistance with zebrafish husbandry and line maintenance. We thank Sourabh Monnappa for his help with PEG-gel preparation. We also thank Petr Strnad and Andrea Boni from Viventis Microscopy for their support with the system and Camille Remy for her help with the UV laser ablation integration to the light sheet microscope.

## Author contributions

E.O., A.C.O. and M.S.S. designed the experiments, E.O. performed the experiments and analyzed the data, E.O., E.M. and M.R. developed and programmed the robotic manipulation system, E.O., A.C.O. and M.S.S. wrote the manuscript, A.C.O. and M.S.S. supervised the research.

## Competing interests

The authors declare no competing interests.
