## [Peer Review File · Nature Communications]

Deconstructing body axis morphogenesis in zebrafish embryos using robot-assisted tissue micromanipulationREVIEWER COMMENTS

Reviewer #1 (Remarks to the Author):

Review of:

Deconstructing body axis morphogenesis in zebrafish embryos using robot-assisted tissue micromanipulation

This manuscript is a frustration, as the goals are so laudable, and the potential impacts are so great, but the presentation of the key aspects of reproducibility and other details are lacking. I urge the editors to help the authors generate the manuscript that is worthy of the potential impact of this work.

Embryonic microsurgery has been a major tool in developmental biology, but it seems to be employed less in recent years, in part because few workers are doing the "apprenticeship" required to master the demanding surgical techniques, and in part because the experimental systems being deployed for many studies were selected because of genetic or other advantages. The manuscript makes these points but not as clearly or compellingly as it might.

This manuscript offers a potential path forward to performing microsurgical manipulations in species that are more challenging or with a faster learning curve: robotic surgery. In clinical settings robotic surgery has impacted practice by permitting finer manipulations, with safe-guards in place. Companies are now developing surgical assist devices that permit the impossible to become the routine in a number of demanding settings such as retinal surgery. That seems like the promise of this manuscript, which is why I am so supportive of its goals.

Why I am so frustrated and why it has taken me far too long to finish this review is that the rigorous demonstration of awesome manipulations and their reproducibility were key parts of winning acceptance by the field and FDA approval. To do so the surgeons selected some high-value demonstrations of the technology. The importance of a biological question, the documentation with rigor and the creation of awesome grafting precision is not really offered in this manuscript (and it must be)

There are many examples of the sloppiness of logic and presentation.
The results needs to present the results, not the conclusions with some of the results.

Here are just a few examples:

On lines 83/84

"To avoid this, we used microscissors attached to the robot to repeatedly produce viable tail explants with controlled size.

This led me to expect a clear documentation of the reproducibility of the explants.
If it is easy to do so with the robotic device, I'd expect the authors to show me the variance across 100 samples, but they didn't

On lines 98/99

"Consistent with measurements performed in intact embryos, the tail explants contained 21 ± 1 somites at the end of somitogenesis ($N = 4$, $n = 12$).

But the control value and the statistical test of how consistent is not offered

The results are often given indirectly, as percentage changes, or comparisons with values not offered, and with significant figures that out-strip the N.

On lines 108-10

However, there was a decrease in the average elongation rate of tail explants by 18% (0.7 ± 0.06 $\mu\text{m}/\text{min}$) when compared to intact embryos (0.9 ± 0.06 $\mu\text{m}/\text{min}$) (Fig. 2b).

This is a value based on $n = 9$ embryos, 13 free explants, 7 embedded explants.

On lines 163/4 0.11 ± 0.02 $\mu\text{m}\cdot\text{min}^{-1}$ ($n = 6$)

On lines 222 71.7 ± 9.4 min ($N = 3$, $n = 6$) <<3 significant figures from $N=3$?>>

On Lines 231/2

Analogous to the results of the notochord ablation experiments, tail elongation ceased for 111.7 ± 24.7 min ($N = 2$, $n = 4$)

<<4 significant figures?>>

These are studies that deserve real numbers that are reported completely and tested for significance. It is just not possible to critically evaluate the claims, or to understand (or believe) the percentages on page 11 (90% 20% 38% 22%)

A side note:

The authors use agarose to hold the specimens – perhaps the most common, but the most sloppy way to mount zebrafish, as the water film between the embryo/larva and the gelled agarose is like a frictionless gimble mount. It is not the way to perform or to document precise manipulations.

Let me completely agree with the authors in the paragraph starting on line 351:

The micromanipulation tools that are introduced in this study enable the precise and reliable mechanical perturbation of the local microenvironment in living embryos, as well as the removal of precisely sized and shaped explants. Analysis of these operations can help to answer outstanding questions regarding whether body axis morphogenesis, or other accessible patterning processes, are best described as the sum of cell-autonomous events, or as phenomena that emerge collectively at the multicellular level.

Now please generate the manuscript that documents the findings, and backs up the claim of “precise and reliable”

I want to love this study and the doors that it opens

Reviewer #2 (Remarks to the Author):

The manuscript by Ozelci, Sakar, and colleagues introduces a robotics-based microsurgical approach for embryology experiments, then illustrates its utility through a series of embryological manipulations toward understanding the mechanisms of axis elongation and somitogenesis in zebrafish. The manuscript is clearly written and well organized, and introduces an approach that could be of tremendous utility across many realms of developmental biology. Overall, therefore, this reviewer is supportive of publication of this manuscript. However, there are some minor concerns that should be addressed to strengthen this very good work further.

1. While the value of the robot-assisted micromanipulation method is generally recognized from the experiments performed, how precise and how advantageous the method is compared to traditional 'manual' embryology is not sufficiently explored. Some quantitative comparison to reveal whether this method is equivalent to a skilled embryologist, or better than one, would be valuable, particularly for readers that do not work with zebrafish embryos and may therefore lack the context to judge these things qualitatively from the data as it is provided.
2. The experiments showing that the period of somitogenesis can be decoupled from elongation are very interesting, but interpretation of these results is not fully fleshed out. Some additional discussion placing these findings in the context of the clock and wavefront model and its more recent variations (e.g. work of Alhuela and colleagues) would be useful in trying to make sense of how these two properties can be decoupled.
3. The statement "Taken together, these data show that the period of the segmentation clock can be decoupled from the biomechanics of elongation, however somite shape and PSM length cannot." is misleading, as it implies the latter is simply not possible when the experiment rather shows that for this particular perturbation they *were not* decoupled.
4. Similarly, the contention that authors show "the notochord elongates independently of tail elongation" is misleading. Authors show that in the absence of tail elongation the notochord does elongate. However, they do not show that it elongates at the same rate as it would in control embryos. Indeed the wildtype tail elongation rate seems to be several fold higher than the notochord elongation rate when tail elongation is disrupted. In other words a significant role for tail elongation in notochord elongation cannot be ruled out unless notochord elongation has been measured in control embryos and is similar to those in which tail elongation is blocked.
5. The term "gripping" forces is also misleading, as it implies something about the nature of forces acting on the notochord that is not yet supported by data (namely that there are large shear traction forces on the notochord that are responsible for maintaining its stability during growth).

Reviewer #1 (Remarks to the Author):

General Comments:

This manuscript is a frustration, as the goals are so laudable, and the potential impacts are so great, but the presentation of the key aspects of reproducibility and other details are lacking. I urge the editors to help the authors generate the manuscript that is worthy of the potential impact of this work. Embryonic microsurgery has been a major tool in developmental biology, but it seems to be employed less in recent years, in part because few workers are doing the “apprenticeship” required to master the demanding surgical techniques, and in part because the experimental systems being deployed for many studies were selected because of genetic or other advantages. The manuscript makes these points but not as clearly or compellingly as it might.

This manuscript offers a potential path forward to performing microsurgical manipulations in species that are more challenging or with a faster learning curve: robotic surgery. In clinical settings robotic surgery has impacted practice by permitting finer manipulations, with safe-guards in place. Companies are now developing surgical assist devices that permit the impossible to become the routine in a number of demanding settings such as retinal surgery. That seems like the promise of this manuscript, which is why I am so supportive of its goals.

Why I am so frustrated and why it has taken me far too long to finish this review is that the rigorous demonstration of awesome manipulations and their reproducibility were key parts of winning acceptance by the field and FDA approval. To do so the surgeons selected some high-value demonstrations of the technology. The importance of a biological question, the documentation with rigor and the creation of awesome grafting precision is not really offered in this manuscript (and it must be).

Let me completely agree with the authors in the paragraph starting on line 351: “The micromanipulation tools that are introduced in this study enable the precise and reliable mechanical perturbation of the local microenvironment in living embryos, as well as the removal of precisely sized and shaped explants. Analysis of these operations can help to answer outstanding questions regarding whether body axis morphogenesis, or other accessible patterning processes, are best described as the sum of cell-autonomous events, or as phenomena that emerge collectively at the multicellular level.” Now please generate the manuscript that documents the findings, and backs up the claim of “precise and reliable” I want to love this study and the doors that it opens.

We thank the reviewer for highlighting the potential of our work and raising important concerns about reproducibility. We hope that the additional data and textual clarifications provided in the revised manuscript will address these concerns, and convince the reviewer that the presented tools enable precise and reliable mechanical perturbation of living embryos.

Specific Comments:

1. There are many examples of the sloppiness of logic and presentation. The results need to present the results, not the conclusions with some of the results.

- On lines 83/84: "To avoid this, we used microscissors attached to the robot to repeatedly produce viable tail explants with controlled size. This led me to expect a clear documentation of the reproducibility of the explants. If it is easy to do so with the robotic device, I'd expect the authors to show me the variance across 100 samples, but they didn't.

We agree with the reviewer that the sample size should have been significantly larger. We designed a new experiment to specifically support our argument on reproducibly producing viable tail explants with controlled size. We added a new supplementary note to describe the experiment in detail (**Supplementary Note 2**) along with an accompanying figure (**Supplementary Figure 22**). The figure is shown below for the convenience of the reviewer.

Supplementary Fig. 22 Experimental study to assess manual and robot-assisted surgery. **a** Schematic illustration summarizing the task. **b** Lateral and dorsal views of a 15-somite stage embryo. Red rectangles indicate the planned location of the cut. Dashed black line shown on the dorsal contour the direction of the cut. Surgery is done by placing the embryo in dorsal view and playing with the focal plane until the target location is in sharp contrast. Scale bars, 100 μm . **c** A scene from the manual microsurgery showing the operator with microscissors and forceps. **d** A scene from the robot-assisted microsurgery with the microscissors mounted on the robot. Unlike the manual microsurgery, the visual is provided by the computer screen, and not the binoculars of the stereomicroscope.

The task is to cut the tail of zebrafish embryos at 15-somite stage in a way that exactly 5 somites would be retained within the tail explant. The manual surgeries were done by a PhD student who has been

performing such surgeries during the last 3 years (completed more than 6000 operations). The student used the same microscissor for both manual and robot-assisted surgeries. We quantified the length of the explant to assess both reproducibility (from the standard deviation) and accuracy (from the mean compared to the control). Here, the deviation of the average length of the explants from the average length of the intact tails that corresponds to the same posterior part is a measure of accuracy. We have also quantified the time of operation as another metric of performance. In summary:

- It took a total of **2 hours** to produce **114 explants** using robotic microsurgery (i.e., approximately 1 minute per operation). During the same time period, 52 explants could be dissected with manual surgery (i.e., approximately 2.3 minutes per operation). We verified that all the explants were viable and completed somitogenesis in the next 7 to 8 hours. Thus, for a trained individual robot-assisted surgery is significantly faster compared to manual surgery for performing microsurgical cuts.

- The length of the explants obtained with manual surgery and robot-assisted surgery were $679 \pm 57 \mu\text{m}$ ($N = 4$, $n = 52$) and $657 \pm 50 \mu\text{m}$ ($N = 4$, $n = 114$), respectively. Compared to the length of the intact tail, which is $628 \pm 26 \mu\text{m}$ ($N = 4$, $n = 101$), explants produced by manual surgery and robot-assisted surgery are 8% and 4.5% longer, respectively. Thus, robot-assisted surgery is more accurate compared to manual surgery in performing microsurgical cuts.

- The standard deviation in data measured from intact embryos is coming from embryo-to-embryo variability during morphogenesis. The standard deviation recorded for robot-assisted microsurgery is higher than the intact tail, but not different between the manual and robot-assisted microsurgery. This is around the size of a single somite ($50 \mu\text{m}$), which is small enough for the investigation we performed in this study. Thus, robot-assisted surgery is as accurate as manual surgery by an experienced individual. This data is presented in a new figure (**Supplementary Fig. 3**) and shown below for the convenience of the reviewer.

Supplementary Fig. 3 Performance analysis of robot-assisted and manual microsurgery. **a** The length of the explant and the tail part that corresponds to the same portion (i.e., starting from the last formed 5 somites until the end of the tail bud) is measured along with the anteroposterior axis following the contour of the notochord. **b** Measured length of the explants obtained with manual ($N = 4$, $n = 52$) and robot-assisted ($N = 4$, $n = 114$) surgery, along with the length of the intact tail that corresponds to the same portion ($N = 4$, $n = 101$).

As a test of the utility for a new-comer to experimental embryology, we have also had an inexperienced undergraduate student with no prior experience in microsurgery complete the same tasks. Within the same time interval (2 hours) the student prepared 25 viable explants using the robot-assisted surgery. Notably, she could not produce a single viable explant with manual surgery. In our future work, we will carefully study the learning process for surgical interventions and program the user interface of the robot to obtain a steep learning curve. As an important observation, younger generation scientists are very familiar with the Xbox gaming controller that we use for teleoperation, which facilitates the use of robotic instruments in biomedical research.

- On lines 98/99: "Consistent with measurements performed in intact embryos, the tail explants contained 21 ± 1 somites at the end of somitogenesis ($N = 4, n = 12$). But the control value and the statistical test of how consistent is not offered.

We were referring to the literature where the number of somites were measured in intact zebrafish embryos (e.g., refs 24-26 in the manuscript). We apologize that we forgot to cite relevant articles to inform the readers. We took this opportunity to generate control data under our laboratory conditions by measuring the number of somites in 30 intact embryos coming from the same line. We measured a total of 30 ± 1 somites ($N = 3, n = 30$) at the end of the somitogenesis process. As explained in the manuscript, while preparing the explants, we discarded the anterior part of the body that contained 10 somites to generate explants with 5 somites. Subtracting 10 from the total number of somites in control samples give 20 ± 1 somites. All measurements are tested for significance using Tukey's paired comparison test.

2. The results are often given indirectly, as percentage changes, or comparisons with values not offered, and with significant figures that out-strip the N. These are studies that deserve real numbers that are reported completely and tested for significance.

In the revised manuscript, all the results are given directly as measured values. In several places in the original manuscript, we only reported the total number of embryos or explants used in the study but forgot to report the number of independent trials (experiments done on different days with embryos coming from different parents). We now report these N values for all our studies. All measurements are tested for significance using Tukey's paired comparison test.

- On lines 108-10: However, there was a decrease in the average elongation rate of tail explants by 18% ($0.7 \pm 0.06 \mu\text{m}/\text{min}$) when compared to intact embryos ($0.9 \pm 0.06 \mu\text{m}/\text{min}$) (Fig. 2b). This is a value based on $n = 9$ embryos, 13 free explants, 7 embedded explants.

For this particular experiment, the data is as follows. ($N = 4, n = 9$ embryos; $N = 4, n = 13$ free explants; $N = 3, n = 10$ embedded explants).

- On lines 163/4 $0.11 \pm 0.02 \mu\text{m}\cdot\text{min}^{-1}$ ($n = 6$)

The data is updated as $0.11 \pm 0.02 \mu\text{m}\cdot\text{min}^{-1}$ ($N = 2, n = 6$).

- On lines 222 $71.7 \pm 9.4 \text{ min}$ ($N = 3, n = 6$) <<3 significant figures from $N=3$?>>

We increased the number of repetitions, and updated the reported value as follows: $68 \pm 15 \text{ min}$ ($N = 5, n = 10$).

- On Lines 231/2: Analogous to the results of the notochord ablation experiments, tail elongation ceased for 111.7 ± 24.7 min (N = 2, n = 4) <<4 significant figures?>>

We increased the number of repetitions, and updated the reported value as follows: 111 ± 30 min (N = 5, n = 11).

- It is just not possible to critically evaluate the claims, or to understand (or believe) the percentages on page 11 (90% 20% 38% 22%).

We increased the number of repetitions, and reported the elongation rates that correspond to each case along with the updated percentage rates. We performed Tukey's multiple comparison test to quantify the significance of differences among the elongation rates. Here is the updated data:

Removal of the entire tailbud dramatically reduced elongation of the remaining explant by 86% (0.1 ± 0.05 $\mu\text{m}/\text{min}$), including that of the notochord (Fig. 5e).

The tail explant continued to elongate with a 22% reduced rate (0.58 ± 0.1 $\mu\text{m}/\text{min}$) compared to the unoperated explant.

Despite the rest of the tissue remaining intact, ablation of the progenitor domain in a diameter of ~ 60 μm reduced the elongation of the tail explant by 48% (0.38 ± 0.1 $\mu\text{m}/\text{min}$), with the posterior end of the notochord always bending ventrally within the tail (Fig. 5c, Supplementary Fig. 17a).

Ablating the same volume of tissue (Supplementary Fig. 18) within the tailbud in a region outside the progenitor domain had a lesser impact, reducing elongation by 29% (0.52 ± 0.1 $\mu\text{m}/\text{min}$) (Fig. 5d, Supplementary Fig. 17b).

4. A side note: The authors use agarose to hold the specimens – perhaps the most common, but the most sloppy way to mount zebrafish, as the water film between the embryo/larva and the gelled agarose is like a frictionless gimble mount. It is not the way to perform or to document precise manipulations.

We appreciate the reviewer's concern about the precision of the confinement technique. As we will explain in the following paragraph, we could not come up with a better technique for various reasons. If the reviewer has a particular suggestion, we would be happy to try.

For the explant to survive during the confinement, it has to be in contact with water. Zebrafish is an aquatic animal that does not have a waterproof skin. Moreover, even mammalian embryos do not survive outside a physiological fluid. For this reason, confinement inside a hydrogel is meaningful. It is true that the space in which the explant resides may act as a frictionless gimble. Nevertheless, during the 3 hours of confinement, the explants neither translated nor rotated. As an alternative to agarose, we embedded the explants inside 5% polyethylene glycol (PEG) gels. PEG gels are soft and hydrated, mimicking the basic physical properties of natural hydrogels without contributing any biochemical signals. As a result, they are extensively used in cell and tissue mechanobiology studies (e.g., Gjorevski et al., *Nature*, **539**:560-564, 2016). The PEG gel was synthesized following a published report (Lee et al., *Acta Biomater*, **10**:4167-4174, 2014). Briefly, 12.5 μl of 4-Arm PEG-thiol (Creative PEGWorks, MW 10k), 12.5 μl of 4-Arm PEG-norbornene (Creative PEGWorks, MW 10k), 5 μl of Lithium phenyl-2,4,6-trimethylbenzoylphosphinate (LAP) photoinitiator (TCI America) and 70 μl of M9 were mixed to synthesize 100 μl of a 5% (v/v) PEG gel. This hydrogel solution was put on explants covered with minimal amount of Leibovitz's L15 medium (ThermoFisher, Catalog No. 21083027) in a glass bottomed

petri dish (35 mm). The specimen was exposed to UV light (340 nm) for 30 seconds to photopolymerize the pre-polymer. After solidification of PEG, 2 ml of L15 medium with Pen-Strep (P/S) (1:200 v/v) and tricaine (1:33 v/v) was added to the petri dish to prevent the explant from drying.

We did not observe any significant change in the morphogenesis of the explants compared to the agarose confinement. As observed in agarose, the elongation rate of the explant was severely reduced and the notochord buckled (see Figure R1 below).

Figure R1: Time-lapse images of a tail explant embedded in 5% PEG. Scale bar, 100 μ m.

An alternative to embedding inside a gel would be to physically compress or constrain the explants using microfabricated devices or apply aspiration with microcapillaries or microfluidic channels. However, mechanical loading of the explant might influence the morphogenesis of internal tissues, an unintended effect that would obscure the results of our study. Moreover, we have to provide the confined explant continuous access to sufficient amount of buffered solution, which is challenging within a microengineered construct. Therefore, we refrained from using such techniques.

Reviewer #2 (Remarks to the Author):

General Comments:

The manuscript by Ozelci, Sakar, and colleagues introduces a robotics-based microsurgical approach for embryology experiments, then illustrates its utility through a series of embryological manipulations toward understanding the mechanisms of axis elongation and somitogenesis in zebrafish. The manuscript is clearly written and well organized, and introduces an approach that could be of tremendous utility across many realms of developmental biology. Overall, therefore, this reviewer is supportive of publication of this manuscript. However, there are some minor concerns that should be addressed to strengthen this very good work further.

We thank the reviewer for his/her motivating and supportive comments.

1. While the value of the robot-assisted micromanipulation method is generally recognized from the experiments performed, how precise and how advantageous the method is compared to traditional 'manual' embryology is not sufficiently explored. Some quantitative comparison to reveal whether this method is equivalent to a skilled embryologist, or better than one, would be valuable, particularly for readers that do not work with zebrafish embryos and may therefore lack the context to judge these things qualitatively from the data as it is provided.

We precisely addressed this comment while answering the first comment of the first reviewer. We refer the reviewer to our reply on Page 2 and 3. We added a new Supplementary Note along with Supplementary Figure 3 and 22 to incorporate this data to the manuscript.

2. The experiments showing that the period of somitogenesis can be decoupled from elongation are very interesting, but interpretation of these results is not fully fleshed out. Some additional discussion placing these findings in the context of the clock and wavefront model and its more recent variations (e.g. work of Alhuela and colleagues) would be useful in trying to make sense of how these two properties can be decoupled.

We thank the reviewer for this important comment. We added a new paragraph in Discussion that includes an interpretation of the reported results in this context which is as follows:

As discussed above, in the freely elongating tail explant, or the embryo⁴⁵, the period of segmentation is balanced with the elongation rate. However, although the period of the segmentation clock is not affected by embedding, tail elongation is effectively stopped. This shows that the period of the clock can be uncoupled from the elongation rate of the tail. In most models of segmentation, the readout of the clock in the anterior PSM is controlled by FGF and Wnt signal gradients that extend anteriorly from the site of synthesis in the tailbud⁴⁸⁻⁵³. The posterior regression of the tailbud in the extending body axis is thus thought to create a co-moving wavefront of signal activity that defines a position in the tissue where the clock's timing is used to determine the segment length. However, the embedding results clearly show that an ongoing posterior regression of signal gradients is not required for the formation of new segments, nor is it required for their correct timing, at least over the 3 hours of our time-lapse

observations. This suggests that the behaviour of segmentation clock cells in the anterior PSM is unlikely to be directly controlled by a threshold - or some other simple feature - of a moving signal gradient²⁹. Nevertheless, it leaves open the possibility that changes in signal levels in the posterior PSM may play a role, potentially by causing changes to the cells leaving the tailbud that become visible with some substantial time delay⁵³.

3. The statement “Taken together, these data show that the period of the segmentation clock can be decoupled from the biomechanics of elongation, however somite shape and PSM length cannot.” is misleading, as it implies the latter is simply not possible when the experiment rather shows that for this particular perturbation they *were not* decoupled.

In this statement, we aimed to convey the message that in a non-elongating explant, the segmentation clock works as it works in a free explant. The somites do form yet due to the limited space available in a non-elongating tail, they are malformed. Hence, confinement leads to changes in the morphology of the somites and the PSM length. We agree with the reviewer’s comment that the way we formulated this sentence may lead to misunderstandings. We revised this sentence as follows:

Taken together, these data show that the period of the segmentation clock can be decoupled from the biomechanics of elongation.

4. Similarly, the contention that authors show “the notochord elongates independently of tail elongation” is misleading. Authors show that in the absence of tail elongation the notochord does elongate. However, they do not show that it elongates at the same rate as it would in control embryos. Indeed the wildtype tail elongation rate seems to be several fold higher than the notochord elongation rate when tail elongation is disrupted. In other words a significant role for tail elongation in notochord elongation cannot be ruled out unless notochord elongation has been measured in control embryos and is similar to those in which tail elongation is blocked.

We apologize for the misleading statement. We measured the elongation rate of the notochord in free explants (controls) and incorporated this data as a third column to the box plot shown in Fig.2i. Indeed, notochord elongation rate is significantly reduced upon confinement. We added the following sentences to the relevant section:

However, elongation rate of the notochord in embedded explants is significantly smaller than the free explants (Fig. 2i). This can be explained by not having enough space to elongate in the embedded configuration.

We also revised the last sentence of the paragraph as follows:

Combined, we show that when the notochord elongates not in tandem with the tail, it does generate enough force from the posterior to buckle. This result suggests that coordination of notochord and tail elongation protects the notochord from mechanical defects.

5. The term “gripping” forces is also misleading, as it implies something about the nature of forces acting on the notochord that is not yet supported by data (namely that there are large shear traction forces on the notochord that are responsible for maintaining its stability during growth).

As mentioned in the manuscript, the term “gripping” actually embraces three different mechanisms that prevent the notochord from sliding backwards: friction, compression, and interlocking. With the existing data, we cannot claim that there are large shear forces acting between the PSM and notochord.

To avoid a potential misunderstanding, we revised the relevant part as follows:

Linking our state-of-the-art experimental embryology results to existing biomechanical theories allows us to speculate that the forces holding the posterior notochord in place during elongation could be generated by three mechanisms: friction, compression, and interlocking due to a phase transition. As reported for universal soft grippers, these are not mutually exclusive mechanisms, and could operate in combination³⁸.

REVIEWERS' COMMENTS

Reviewer #1 (Remarks to the Author):

I appreciate the care with which the revision was created, and the clarity that has resulted from reporting the numbers, the values, and the sources of the comparison measurements. I commend the authors for their efforts in making this a meaningful contribution to the literature.

My comments about agarose were not intended as a call to raise the explants in dry conditions, but to contain them in a different gel, as they have. In our own work, gels such as polyacrylamide have provided better and more consistent containment. The authors' clarification and alternative gel answer my concerns

I have two small details to mention:

1 - the abbreviation NC is a bit confusing for many readers as it is the abbreviation typically used for neural crest. Other authors have avoided this potential confusion with abbreviations such as Noto or No, and I hope the authors will consider an alternative.

2 - the imaging and assistance with/using Viventis Microscopy is excellent. I wonder if there is enough of a linkage of the authors to Viventis that there is a relationship to disclose.

My compliments to the authors

Scott E Fraser

Reviewer #2 (Remarks to the Author):

I am satisfied with the responsive resubmission.

-NLN

Reviewer #1 (Remarks to the Author):

General Comments:

I appreciate the care with which the revision was created, and the clarity that has resulted from reporting the numbers, the values, and the sources of the comparison measurements. I commend the authors for their efforts in making this a meaningful contribution to the literature. My comments about agarose were not intended as a call to raise the explants in dry conditions, but to contain them in a different gel, as they have. In our own work, gels such as polyacrylamide have provided better and more consistent containment. The authors' clarification and alternative gel answer my concerns.

My compliments to the authors

We are glad that the revised manuscript has successfully addressed the reviewer's concerns. We would like to thank for all the insightful comments and suggestions.

Specific comments:

1 - the abbreviation NC is a bit confusing for many readers as it is the abbreviation typically used for neural crest. Other authors have avoided this potential confusion with abbreviations such as Noto or No, and I hope the authors will consider an alternative.

We thank the reviewer for raising this point. In order to avoid potential confusion, we exchanged the abbreviation "NC" with "Noto".

2 - the imaging and assistance with/using Viventis Microscopy is excellent. I wonder if there is enough of a linkage of the authors to Viventis that there is a relationship to disclose.

Authors verify that there is no conflict of interest between the authors and the Viventis company.

Reviewer #2 (Remarks to the Author):

General Comments:

I am satisfied with the responsive resubmission.

We are glad that the revised manuscript has successfully addressed the reviewer's concerns.